# An Analysis of Legislative Support Effect for Circular Economy Development in the Context of "Double Carbon" Goal in China

Ruisi Gao [1], Hongfang Han [2], Xueting Zeng [2,*], Xinyu Zhang [2,*] and Xuejing Yang [3]

1    School of Law, Capital University of Economics and Business, Beijing 100071, China
2    School of Labor Economics, Capital University of Economics and Business, Beijing 100071, China
3    School of Law, Henan University of Technology, Jiaozuo 454003, China
*    Correspondence: zengxueting@cueb.edu.cn (X.Z.); xyzhang@cueb.edu.cn (X.Z.)

**Abstract:** In the requirement of the "double carbon" goal, China has confronted the lack of a driving force for the low-carbon transformation of socioeconomic development, which requires a comprehensive law and strategy support system for supporting a circular economy. In this study, a framework associated with a multi-level quantitative index system associated with legislative support for a circular economy (MILC) through the entropy TOPSIS method was developed. It can not only reflect the legal support for a circular economy in four areas based on the "3R" principle, but also respond to the process of environmental law improvement dynamically. The legislative support effect can be applied and analyzed in China's circular economy for the period from 2009 to 2022, which can respond to the process of legal improvement on the environment dynamically. The obtained results show that China's circular economy and its rule-of-law guarantee system are still facing many challenges, such as the low systematic degree of legalization for industrial development, unmatched supporting legal system, and backward concepts and consciousness of circular economy legalization. Various suggestions according to the identification of the importance of a legal support system for the circular economy were obtained, which can not only encourage a reduction in resource consumption and carbon reduction from the view of systematic legalization, but also promote socioeconomic transformation to match the goal of "double carbon".

**Keywords:** double carbon target; circular economy development; a multi-level quantitative index system; legislative support effect

## 1. Introduction

In 2020, China set the goals of peaking $CO_2$ emissions by 2030 and achieving carbon neutrality by 2060 (called the "double carbon" goal), which requires the Chinese government to take a series of positive actions (including market access, legal improvement, industrial structure adjustment, and consumption pattern optimization) to confront the contradiction between human energy consumption structure and resource and environmental constraints. However, the high proportion of energy-consuming industries, extensive production modes, and the huge energy consumption per unit of output will cause an increase in carbon emissions under rapid economic development. Thus, the establishment of a green low-carbon-cycle economic system should be established, which can not only enhance the efficiency of resource circulation and alleviate the resource constraint problem, but also make the socioeconomic system and the natural ecological system coexist harmoniously.

In the requirement of the "double carbon" goal in China, the circular economy can reduce energy consumption and carbon emissions by extending the life cycle of products and reducing the processing stages of products [1,2]. On the other hand, it can also significantly improve the carbon sequestration capacity of products, which has become an indispensable driving force to achieve the goal of "double carbon". According to "The Economics of the Coming Spaceship Earth" first proposed by Boudling, various research works have been

developed to explore in detail the symbiotic relationship between environmental ecosystems and people through "Circular Its Wasters" and "Circular Flow" [3]. There is extensive research on the application of the circular economy theory regarding "resource output", "rural economic development", "solid waste pollution prevention", and "waste recycling" issues [4–8]. Meanwhile, various quantitative methods for evaluating the circular economy, mainly according to the three methods material flow analysis, eco-efficiency, and ecological footprint, have been proposed to reflect the features and corresponding ecological efficiency. In general, material flow analysis has been proposed to study inflow–outflow material in the production and consumption areas, which can identify the special feature of the growth mode of a circular economy [9–11]; However, it cannot reflect the ecological effects of a circular economy, but can only reflect the corresponding circular economy efficiency at the national/regional level. Thus, the eco-efficiency method is used to evaluate the developing level of a circular economy according to ecological effects, where a number of methods, such as the influencing mechanism analysis, the eco-efficiency metric method, and the efficiency optimal model, have been proposed [12–14]. However, various influences associated with subjective and objective factors considered in the influencing mechanism analysis or the optimal model can increase the complexity of the estimating process of a circular economy. Therefore, the ecological footprint method can be proposed to quantify the natural resources needed to support economic development through an ecological accounting system, which can also measure the human impact on the environment based on the certain mode of economic development (such as a circle manner) [15–17]; however, the conceptualized data through the ecological footprint are not an effective/direct means to support the adjustment of the current policies of a circular economy.

Under these situations, a series of measures (including environmental policy, public awareness, and technological progress) have been developed, which are beneficial in improving the resource output rate and the comprehensive efficiency of a circular economy [18–26]. Among them, legislation construction can promulgate (or make "enacted") the developing mode by a legislature or other governing body, which can facilitate economy development in a "reducing, reusing and recycling" manner [27]. During the 1970s, the rapid growth of the global economy and changes in consumption patterns led to an increasing awareness of the finite nature of resources and the escalating waste issue. It was within this context that the Reduce, Reuse, and Recycle (denoted as "3R") principle emerged. The "3R" principle gained prominence as a response to the environmental challenges of that era. The idea of reducing waste through conscious consumption, reusing materials and products whenever possible, and implementing recycling systems gained traction among environmentalists, policymakers, and the general public [28,29]. Various researchers have focused on the qualitative analysis of legislative promulgation on the circular economy, with aim to achieve the goal of "double carbon" in China [30–33]. Nevertheless, little attention has been paid to the quantitative analysis of the role of legislation and the corresponding supporting or promoting level for the circular economy (Table 1).

This paper aims to develop a multi-level quantitative index system for legislative support in the circular economy (MILC) through the entropy TOPSIS method, which can reflect the dynamic impact of changes from various rules of law based on current policy and regulation application. It can not only express legal support for the circular economy in four areas (including resource output, waste recycling, environmental protection, and economic and social development), but also responds to the process of environmental law improvement dynamically. This MILC was applied for actual development of the rule of law in China's circular economy from 2009 to 2022. The results show that the development of China's circular economy and its rule-of-law guarantee are still facing many challenges, including the low systematic degree of legalization for industrial development, an unmatched supporting legal system, and backward concepts and consciousness of circular economy legalization. Various suggestions according to the identification of the importance of a law support system for a circular economy were obtained, which can not only reduce resource consumption and carbon reduction from the view of systematic

legalization but also promote economic transformation to match the goal of "double carbon" in a legal support and institutional improvement manner.

**Table 1.** The latest research status of circular economy.

| Research Content | | Literature | Perspective of Research | Pros and Cons |
|---|---|---|---|---|
| Quantitative methods for evaluating circular economy | Material flow analysis | [9–11] | Inflow–outflow material in the production and consumption areas | It can identify the special feature of growth mode of circular economy but without its ecological effect. |
| | Eco-efficiency method | [12–14] | The perspective of ecological effect | It can evaluate the developing level of circular economy according to ecological effect but increases the complexity of estimating process. |
| | Ecological footprint | [15–17] | Ecological accounting system | It can measure the human impact on the environment based on the certain mode of economic development, but not a direct measuring way. |
| The application of circular economy theory | | [4–6] | Resource output | It can integrate ecological economy and sustainable development into a framework for research but lacks paradigm of economics. |
| | | [6] | Rural economic development | |
| | | [7] | Solid waste pollution prevention | |
| | | [8] | Waste recycling | |
| Subjective and objective factors affecting circular economy | | [18–20] | Environmental policy | These factors can promote the development of circular economy but lack research on legal factors. |
| | | [21–23] | Public awareness | |
| | | [24–26] | Technological progress | |
| | | [30–33] | Qualitative analysis of legislative promulgation on circular economy | It can analyze and study the essence of the influence of legal factors on circular economy but lacks objectivity and accuracy. |

## 2. Case Study

### 2.1. Problem Statement

The concept of circular economy was first introduced to China in 2002, which was formally established as a development goal with the release of "Opinions on Accelerating the Development of Circular Economy" by the State Council in 2004. China has gradually increased its investment and efforts in promoting the circular economy in recent decades. Particularly, in 2015, the "National Circular Economy Development Outline (2015–2020)" was issued, which outlined specific tasks and goals for the circular economy development (i.e., improving resource utilization efficiency, strengthening waste management, and promoting the development of green industries). These initiatives have driven the development speed of the circular economy and expanded the scale and range of developing mode transformation. Up until 2020, the value of China's circular economy reached more than CNY 12 trillion.

However, under the rapid development of the circular economy, China faces certain challenges, particularly in the realm of institutional and legal safeguards, as follows: (a) The circular economy not only encompasses the aspects of material flow and resource utilization but also involves multiple dimensions, such as economy, society, and the environment. Thus, it leads to the need to simultaneously consider the interplay of these dimensions when assessing the effectiveness and performance of the circular economy, which increases the complexity of evaluation. (b) Although the government has implemented a series of policies and measures to guide enterprises in improving production processes, enhancing resource utilization efficiency, and promoting green consumption, an insufficient driving force for production transformation and consumption intensification

requires further strengthening of effective policies and institutional safeguards to improve the efficiency of these transformations and changes. (c) The proliferation of regional institutional frameworks without a national perspective can reduce the legislative effects, which can be deemed as a hold-back for circular economy development under dual-carbon goals. To foster the synergistic effects of circular economy development across different regions, a unified policy framework and institutional system need to be established nationwide. Furthermore, under the impetus of the "double carbon" goal, the circular economy must synergize with the low-carbon economy to achieve the dual objectives of resource recycling and carbon reduction. Therefore, this article constructs the framework of legislative support for circular economy development in the context of the "double carbon" goal (as shown in Figure 1) to dynamically reflect the legalization degrees of circular economy development and corresponding support effects based on current policy and regulation implementation.

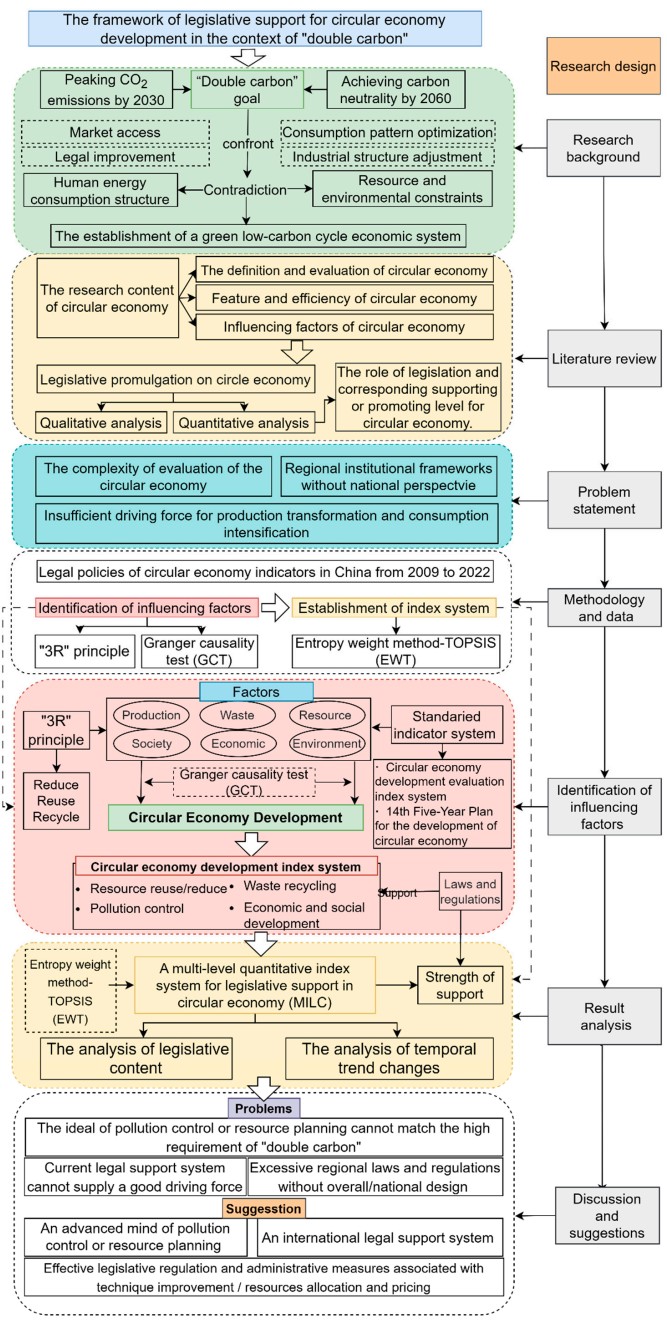

**Figure 1.** The framework of legislative support for circular economy development in the context of "double carbon".

*2.2. Methodology*

2.2.1. Identification of Circular Economy Development Based on "3R" Principle

Based on relevant theoretical and practical research associated with the index construction of circular economy development, many factors (such as production patterns, the resource use model, ecological and environmental state, and socioeconomic benefits) can be considered as key indicators to identify the developing level of a circular economy [34–36]. However, the circular economy is developing rapidly and involving a wide range of fields, which leads to various logics of index system construction. Among them, the "3R" principle can be deemed as a general principle of building an indicator system, which can reflect the basic features of a circular economy (low consumption, low emission, and high efficiency) [35]. The establishment of the Reduce, Reuse, and Recycle (denoted as "3R") principle can be traced back to the 1970s when the concepts of environmental conservation and sustainable development began to gain attention. The "3R" principle, which stands for Reduce, Reuse, and Recycle, originated from waste management and energy recovery from waste [28,29]. Meanwhile, a number of standardized indicator systems induced by China's ministries and commissions (such as in 2017 by four ministries, the Ministry of Finance, Development and Reform Commission, Ministry of Environmental Protection, and National Bureau of Statistics) promulgated the "Circular Economy Development Evaluation Index System"; in 2021, the National Development and Reform Commission promulgated the "14th Five-Year Plan for the development of circular economy", which distinguishes the degree of circular economy development at both macro and micro levels. Under these situations, according to the above standardized index system, resource reuse/reduction, waste recycling, pollution reduction, economic and social development, society and livelihood, and resource and environmental security can be selected as the main considerable factors to calculate legalization degrees of circular economy development. The Granger causality method was used to test if these factors have relationships with the circular economy [37]. Based on variance of the best least squares prediction of all information at some point in the past, it was found that resource reuse/reduction, waste recycling, pollution reduction, and economic and social development were the most important indicators (as shown Table 2).

**Table 2.** Results of Granger's causality test.

| Variable | *p*-Value | Explanation |
|---|---|---|
| Resource reuse/reduction | 0.0000 | Resource reuse/reduction is the Granger cause of circular economy |
| Waste recycling | 0.0000 | Waste recycling is the Granger cause of circular economy |
| Pollution reduction | 0.0071 | Pollution reduction is the Granger cause of circular economy |
| Economic and social development | 0.0000 | Economic and social development are the Granger cause of circular economy |
| Society and livelihood | 0.2174 | Society and livelihood are not the Granger cause of circular economy |
| Resource and environmental security | 0.2345 | Resource and environmental security are not the Granger cause of circular economy |

In addition, with reference to the standards issued by the four ministries and the NDRC [38] and taking into account the actual situation in China, specific indicator items were selected. Thus, a circular economy development based on the "3R" principle was constructed to evaluate the development status of the circular economy (as shown in Figure 2).

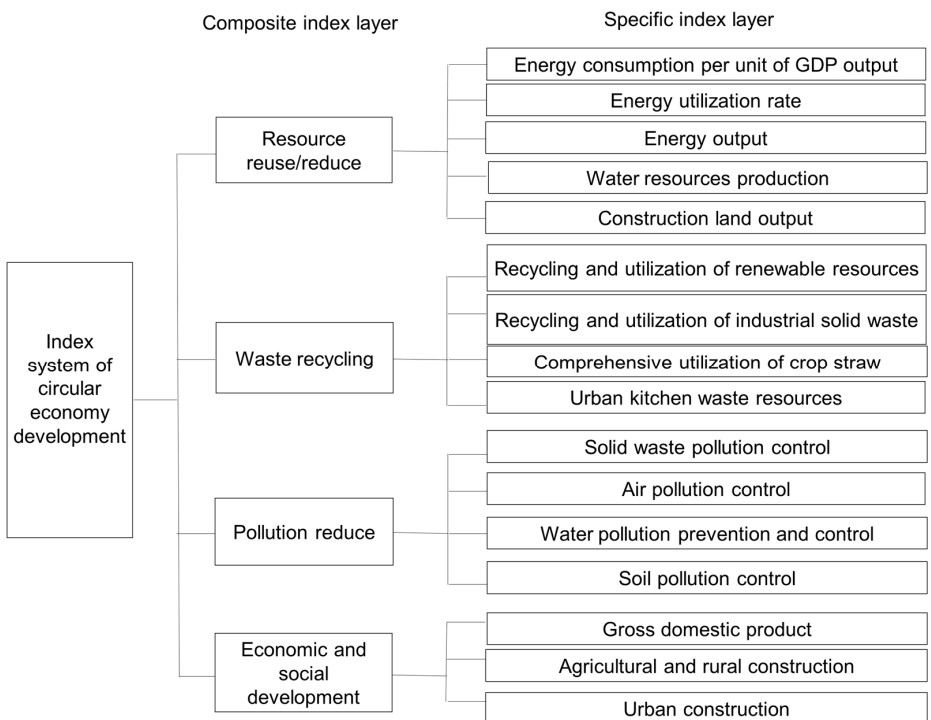

**Figure 2.** Circular economy development index system.

Although evaluating the circular economy through "3R" principles can effectively identify the degree of circular economy development at different levels from multiple perspectives, it is difficult to supply the quantitative analysis for adjusting current policies (especially laws). Therefore, we need a more effective method to reflect the level of legalized support for the circular economy.

### 2.2.2. A Multi-Level Quantitative Index System for Legislative Support in Circular Economy (MILC) through Entropy Weight TOPSIS Method

In general, a legal support system for a circular economy needs to be consistent with three criteria (clear purpose, perfect structure, and ability to implement). Based on the circular economy development index system, a multi-level quantitative index system for legislative support in the circular economy (MILC) was constructed from legislative purpose, legislative content, legislative level, and legislative effect (as shown in Figure 3).

In Figure 3, the legislative aim of laws and regulations is to promote circular economy development from four specific areas (resource reuse, waste recycling, resource reduction, and economic and social development) in the evaluation index system of the circular economy. Since laws and regulations have their own characteristics (special concerns and manners), five aspects of law, administrative regulations, judicial interpretation, departmental regulations, and local legislation can be considered in the MILC. Among them, the law has the strongest authority and the widest scope of effect, while local legislation has relatively weaker authority and a narrower scope of effect; in terms of legislative support effect, laws and regulations are examined according to their applicability to practice (including appropriateness and operability). However, various subjective factors can affect the efficiency of the evaluation of the MILC; thus, an Entropy Weight Technique for Order Preference by Similarity to an Ideal Solution (EWT) was used to assign objective weights to the indicators. In general, the EWT can assign values to different levels of circular economy development on three levels (including hierarchy assignment, fit assignment, and operability assignment), which can increase objectivity and better interpret the results obtained [39]. The EWT was introduced into the MILC to calculate the support of different laws and regulations to the elements of circular economy development as follows:

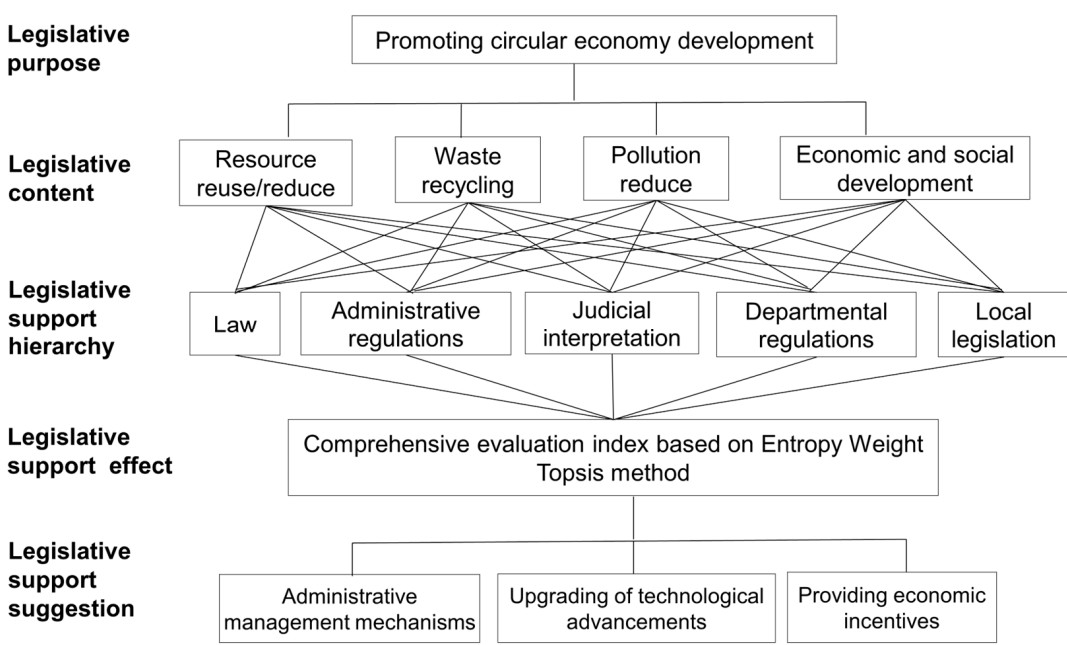

**Figure 3.** A quantitative framework of the rule of law for circular economy development (MILC).

(1) Suppose there are m evaluated objects and *n* evaluation indexes for each evaluated content; then, construct the judgment matrix:

$$X = (x_{ij})_{m \times n} \ (i = 1, 2, \ldots, m; j = 1, 2, \ldots, n) \tag{1}$$

(2) Normalize the judgment matrix:

$$x'_{ij} = \frac{x_{ij}}{x_{max}} \tag{2}$$

Here, $x_{max}$ is the maximum value under the same indicator.

(3) Calculate information entropy:

$$H_j = -k \sum_{i=1}^{m} p_{ij} \, ln p_{ij} \tag{3}$$

where $p_{ij} = \frac{x'_{ij}}{\sum_{i=1}^{m} x'_{ij}}; k = \frac{1}{\ln m}$.

(4) Define the weights of indicator *j*:

$$\omega_j = \frac{1 - H_j}{\sum_{j=1}^{n} (1 - H_j)} \tag{4}$$

where $\omega_j \epsilon [0, 1]$ and $\sum_{j=1}^{n} \omega_j = 1$.

(5) Calculate the weighting matrix:

$$R = (r_{ij})_{m \times n}, r_{ij} = \omega_j \cdot x_{ij} \ (i = 1, 2, \ldots, m; j = 1, 2, \ldots, n) \tag{5}$$

(6) Determine the optimal solution $S_j^+$ and the inferior solution $S_j^-$:

$$S_j^+ = \max(r_{1j}, r_{2j}, \ldots, r_{nj}), \ S_j^- = \min(r_{1j}, r_{2j}, \ldots, r_{nj}) \tag{6}$$

(7) Calculate the Euclidean distance of each solution from the optimal and inferior solutions:

$$sep_i^+ = \sqrt{\sum_{j=1}^{n} \left( S_j^+ - r_{ij} \right)^2}, \quad sep_i^- = \sqrt{\sum_{j=1}^{n} \left( S_j^- - r_{ij} \right)^2} \tag{7}$$

(8) Calculate the composite evaluation index:

$$C_i = \frac{sep_i^-}{sep_i^+ + sep_i^-}, C_i \in [0,1] \tag{8}$$

Here, the larger the A value, the better the evaluation object. Finally, some legislative support solutions of circular economy development are put forward based on the composite evaluation index, which provides support for political advice.

*2.3. Data Analysis*

In terms of data sources, some keywords (including "circular economy development", "resource output", "waste recycling", "solid waste pollution prevention", "rural economic development", etc.) were used based on the legal database to ensure the completeness and credibility of laws and regulations related to circular economy development. A number of laws, regulations, and policy documents related to circular economy development were also taken into account, issued by the National Development and Reform Commission, the Ministry of Industry and Information Technology, the Ministry of Environmental Protection, and other relevant departments [38–41]. The collected policy texts, laws, and regulations were further cross-referenced and screened according to the conceptual scope and implementation of the circular economy. Under such screening principles, 3200 texts at the five levels of law, administrative regulations, judicial interpretation, departmental regulations, and local legislation were finally retained (as shown in Table 3). The levels of enacting bodies of circular economy laws and regulations vary greatly, which could lead to gaps in the strength of regulations. Moreover, a number of policies in local legislation are mainly derived from local regulations, plans, and action plans issued by the governments of provinces, municipalities, and autonomous regions, involving policy initiatives for the development of a circular economy. In addition, data on circular economy development indicators are obtained from annual government work reports and policy documents, as well as the China Statistical Yearbook and China Environmental Statistical Yearbook [42].

**Table 3.** Law policies for circular economy development, 2009–2022.

| Legislative Content | Legislative Level | Number of Policies (2009–2022) | | | | | | | | | | | | | |
|---|---|---|---|---|---|---|---|---|---|---|---|---|---|---|---|
| | | 2009 | 2010 | 2011 | 2012 | 2013 | 2014 | 2015 | 2016 | 2017 | 2018 | 2019 | 2020 | 2021 | 2022 |
| Resource reuse/reduction | Law | 0 | 0 | 1 | 0 | 0 | 0 | 0 | 0 | 0 | 1 | 0 | 0 | 0 | 0 |
| | Administrative regulations | 0 | 0 | 0 | 1 | 3 | 1 | 0 | 1 | 0 | 0 | 0 | 0 | 2 | 1 |
| | Judicial interpretation | 0 | 0 | 0 | 0 | 0 | 0 | 0 | 0 | 0 | 0 | 0 | 0 | 0 | 0 |
| | Departmental regulations | 1 | 6 | 6 | 10 | 7 | 2 | 5 | 8 | 10 | 3 | 1 | 2 | 3 | 0 |
| | Local legislation | 16 | 19 | 61 | 62 | 64 | 70 | 51 | 121 | 105 | 59 | 26 | 21 | 62 | 72 |
| | Total | 17 | 25 | 68 | 73 | 74 | 73 | 56 | 130 | 115 | 63 | 27 | 23 | 67 | 73 |
| Waste recycling | Law | 0 | 0 | 0 | 0 | 0 | 0 | 0 | 1 | 0 | 0 | 0 | 0 | 0 | 0 |
| | Administrative regulations | 5 | 1 | 0 | 0 | 1 | 0 | 0 | 2 | 0 | 0 | 0 | 0 | 0 | 0 |
| | Judicial interpretation | 0 | 0 | 0 | 0 | 0 | 0 | 0 | 0 | 0 | 0 | 0 | 0 | 0 | 0 |
| | Departmental regulations | 25 | 2 | 2 | 3 | 5 | 3 | 11 | 7 | 8 | 10 | 3 | 2 | 7 | 5 |
| | Local legislation | 30 | 25 | 51 | 53 | 40 | 43 | 50 | 97 | 117 | 96 | 57 | 41 | 62 | 82 |
| | Total | 293 | 28 | 53 | 56 | 46 | 46 | 61 | 107 | 125 | 106 | 60 | 43 | 69 | 87 |

**Table 3.** *Cont.*

| Legislative Content | Legislative Level | Number of Policies (2009–2022) | | | | | | | | | | | | | |
| | | 2009 | 2010 | 2011 | 2012 | 2013 | 2014 | 2015 | 2016 | 2017 | 2018 | 2019 | 2020 | 2021 | 2022 |
| Pollution reduction | Law | 1 | 6 | 4 | 3 | 4 | 2 | 3 | 4 | 3 | 5 | 1 | 1 | 3 | 0 |
| | Administrative regulations | 2 | 11 | 13 | 14 | 13 | 8 | 9 | 9 | 8 | 2 | 2 | 0 | 2 | 2 |
| | Judicial interpretation | 0 | 1 | 0 | 0 | 0 | 1 | 0 | 1 | 1 | 0 | 0 | 0 | 0 | 2 |
| | Departmental regulations | 20 | 31 | 57 | 63 | 45 | 29 | 54 | 43 | 47 | 23 | 20 | 24 | 19 | 13 |
| | Local legislation | 209 | 284 | 200 | 204 | 139 | 191 | 215 | 256 | 332 | 163 | 100 | 113 | 112 | 53 |
| | Total | 232 | 333 | 274 | 284 | 201 | 231 | 281 | 313 | 391 | 193 | 123 | 138 | 136 | 70 |
| Economic and social development | Law | 4 | 6 | 4 | 3 | 4 | 2 | 2 | 4 | 3 | 3 | 1 | 0 | 2 | 0 |
| | Administrative regulations | 12 | 10 | 17 | 12 | 8 | 5 | 5 | 7 | 3 | 2 | 0 | 0 | 3 | 1 |
| | Judicial interpretation | 0 | 0 | 0 | 0 | 0 | 1 | 0 | 0 | 0 | 0 | 0 | 0 | 0 | 0 |
| | Departmental regulations | 22 | 30 | 39 | 39 | 19 | 15 | 23 | 18 | 23 | 9 | 5 | 4 | 8 | 3 |
| | Local legislation | 5 | 7 | 10 | 14 | 6 | 11 | 7 | 9 | 22 | 28 | 24 | 26 | 31 | 11 |
| | Total | 43 | 53 | 70 | 68 | 37 | 34 | 37 | 38 | 51 | 42 | 30 | 30 | 44 | 15 |

Table 4 presents the weights (R) and optimal and worst solutions ($S_j^+$ and $S_j^-$) for the legislative indicators based on the EWT. The larger difference between the optimal solution $S_j^+$ and worst solution $S_j^-$ for the legislative level indicator can be observed, where the greater the variation, the smaller the entropy value. Conversely, indicators with larger entropy values have smaller entropy weights. In the weights of indicators, pollution reduction holds the highest weight among the legislative levels (>0.1), followed by economic and social development. On the other hand, resource reuse/reduction and waste recycling have relatively smaller weighted weights across the legislative levels. Furthermore, there is a significant difference between the optimal solution ($S_j^+$) and the worst solution ($S_j^-$) for administrative regulations and local legislation in the legislative-level indicators. This suggests a higher degree of variation, which indicates that these legislative levels are significantly better than other legislative levels across multiple indicators.

**Table 4.** The weighted weight, optimal solution, and worst solution of each indicator.

| | Law | Administrative Regulations | Judicial Interpretation | Departmental Regulations | Local Legislation |
| --- | --- | --- | --- | --- | --- |
| Resource reuse/reduction | 0.0103 | 0.0316 | 0 | 0.0376 | 0 |
| Waste recycling | 0 | 0 | 0 | 0.0418 | 0.0080 |
| Pollution reduction | 0.1035 | 0.2687 | 0.2268 | 0.1015 | 0.2578 |
| Economic and social development | 0.1449 | 0 | 0.0567 | 0 | 0.0198 |
| Optimal solution ($S_j^+$) | 0.1449 | 0.2687 | 0.2268 | 0.1015 | 0.2578 |
| Worst solution ($S_j^-$) | 0 | 0 | 0 | 0 | 0 |

Table 5 calculates the comprehensive scores ($C_i$) of the county-level economy according to the EWT method. The results showed that a smaller distance ($sep_i^+$) from the optimal solution ($S_j^+$) indicates better competitiveness at the county level, while a larger distance ($sep_i^-$) from the worst solution ($S_j^-$) indicates better performance. Within the legislative evaluation indicator system, the comprehensive evaluation values for the four legislative components (i.e., resource reuse/reduction, waste recycling, pollution reduction, and economic and social development) are 0.0127, 0.0085, 0.9919, and 0.1278. It can be inferred that these four legislative components play a crucial role in the development of the circular economy.

**Table 5.** Comprehensive evaluation score of the legislative system.

|  | $Sep_i^-$ | $Sep_i^+$ | $C_i$ |
|---|---|---|---|
| Resource reuse/reduction | 0.1963 | 0.0025 | 0.0127 |
| Waste recycling | 0.2106 | 0.0018 | 0.0085 |
| Pollution reduction | 0.0017 | 0.2112 | 0.9919 |
| Economic and social development | 0.1681 | 0.0246 | 0.1278 |

Figure 4 shows the data description section and methodology section, which could illustrate the way to collect and organize the sources of data and the methods and principles of processing and analyzing these data.

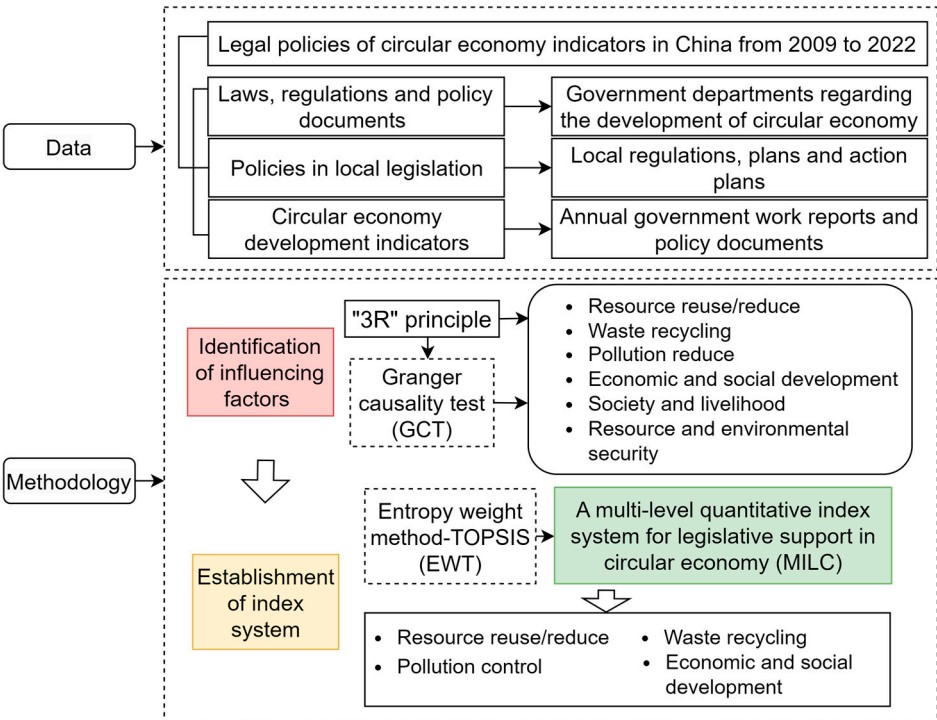

**Figure 4.** Framework of data description and model development.

## 3. Results and Discussion

### 3.1. Results Analysis

According to quantitative results through the MILC, various results associated with the total legislative support effect for circular economy development were obtained, which can reveal the development pattern of China's circular economy and the features of legislative support for the circular economy dynamically. It was found that the total legislative support effect for circular economy development was expressed as dynamic changes, where the highest level occurred in 2017 and the lowest level appeared in 2020. Meanwhile, the four corresponding legislative content and legislative hierarchy components present different roles in legal support effects from 2009 to 2022.

3.1.1. The Analysis of Temporal Trend Changes in Total Legislation Support Effect and the Legislation Support Effect of Each Legislative Area for Circular Economy Development

Figure 5 shows the change in the legislative support effect of the circular economy in China from 2009 to 2022, reflecting the variation trend of declining to rising to declining. From 2009 to 2017, the policy quantity increased from 585 to 682, and the supporting effect increased from 0.58 to 0.64, which shows that the number of policies issued by the Chinese government to promote the development of circular economy and the overall level of policy support are constantly improving. Although there was significant fluctuation in

2010 and 2013, the overall trend showed a clear increase with a peak of support effect by 0.64 in 2017, which indicates that the Chinese government's attention to promoting the development of the circular economy has been increasing with the country's growing economic strength and the worsening global climate. However, the number of policies issued annually decreased significantly after the introduction of a large number of policies in 2017, especially local regulations, which resulted in a decrease in policy support level. This indicates that China's legislative concepts and legal system design for the circular economy still lack systematicity compared to developed countries. In fact, systematicity refers to the interconnectedness and coordination among laws and regulations, which can be incorporated into a legislative support system for reflecting the synergy level between the legislative content and legislative hierarchy for the circular economy.

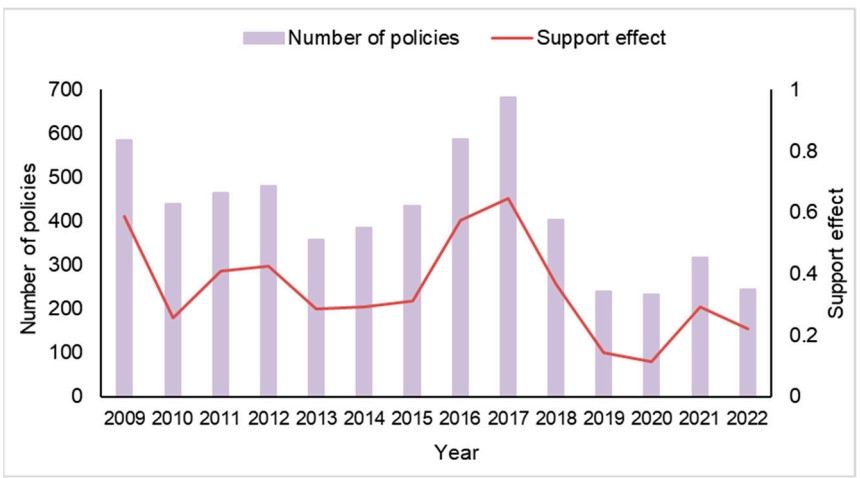

**Figure 5.** The change in policies and support effect for the legal development of the circular economy in China from 2009 to 2022.

Figure 6 displays the change in the support effect under four areas (i.e., resource reuse/reduction (RR), waste recycling (WR), pollution reduction (PR), and economic and social development (ESD)) for the legal development of the circular economy in China from 2009 to 2022. The results are as follows: (a) The overall support effect trend of the four legislative components in the legal institutionalization of China's circular economy development shows an initial increase followed by a decrease, and the development gap between them is gradually narrowing. It can be observed that the legislative area of pollutant reduction has been consistently leading among the other three areas (i.e., RR, WR, and ESD), which reflects that waste reduction has long been the primary means of legal institutionalization in China's circular economy development. (b) The legislative support effects for RR, PR, and ESD showed an upward trend in 2009. In particular, the legislative area of PR increased rapidly, reaching its peak in 2010 with a support effect of 0.64. In contrast, WR showed a downward trend, WR showed a downward trend, with a WR of 0.092 in 2009 and 0.0186 in 2010. The results reflect that the Chinese government paid particular attention to pollution reduction at that time to reduce the serious negative impact of environmental pollution on public health and sustainable development. (c) Subsequently, the development gap among the four legislative areas showed an overall narrowing trend from 2010 to 2021. Specifically, all four legislative areas showed a downward trend since 2016, while RR showed an upward trend, and its support intensity surpassed WR legislation. Additionally, the legislative support effect for pollutant reduction rapidly increased after 2021, reaching a support effect of 0.52 in 2022, which reflects that the legal institutionalization of China's circular economy development has gradually relied more on pollutant reduction legislation.

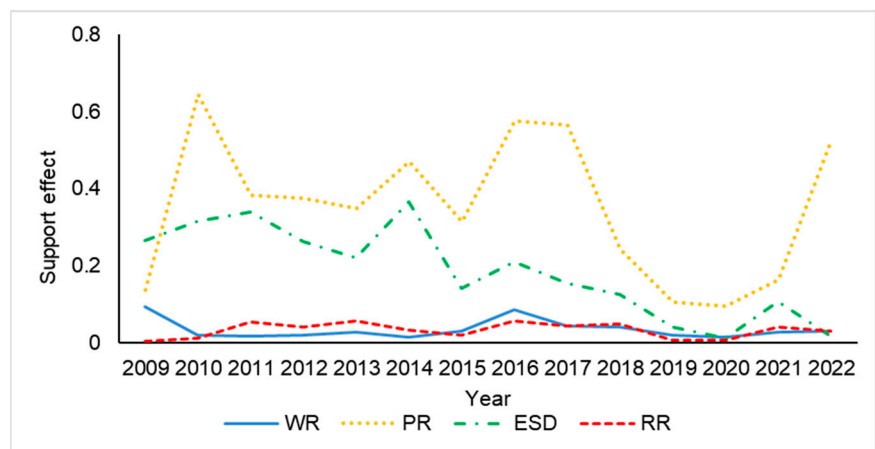

**Figure 6.** The change in the support effect under four legislative areas from 2009 to 2022 (RR—resource reuse/reduction; WR—waste recycling; PR—pollution reduction; ESD—economic and social development).

### 3.1.2. The Analysis of Legislative Content

Figure 7 shows the total support effect of legislative content and the corresponding legislation hierarchy, illustrating that a higher level of support for legislative content does not necessarily imply that each corresponding legislative hierarchy is well-developed. The obtained results are as follows: (a) Compared to other legislative areas, pollution reduction (PR) has the highest support effect of all the types of legislation. In comparison to resource reuse/reduction (RR), waste recycling (WR), and economic and social development (ESD), the highest support effect of PR content reached 0.95, which demonstrates that in the process of constructing legislation in the circular economy, the government focuses more on environmental protection but ignores issues such as resource consumption patterns and recycling methods. When the problem occurs, the legislative ideal of solving problems is not conducive for sustainable resource/environment management in the long term. (b) In PR, the support effect of local legislation (LL) is as high as 0.258, which indicates that China attaches more importance to local legislation to promote the regional development level of the circular economy. Excessive strong targeting of local legislation can solve resource waste and pollution problems at a small scale, but this loses the overall picture for a national objective. In the long term, it can easily lead to an increase in regional legal heterogeneity, resulting in uneven circular economy development in various regions of China. (c) Resource reuse/reduction (RR) and waste recycling (WR) can be mainly supported by departmental regulations, which have lower regulatory strength. The total policy support effect of DR in the legislative areas of RR and WR reached 0.38 and 0.41, respectively, which implies that relevant regulation and legislative documents have been issued in a timely manner and that measures have been taken by various departments. For example, in order to promote the recycling of renewable resources, the National Development and Reform Commission issued the "Regulations on the Management of Renewable Resource Recycling" as a departmental regulation. However, the regulatory strength of these regulations is relatively weak and lacks systematicity, which indicates that there is a lack of interconnection and coordination among these laws and regulations, and the legal provisions lack a rational and clear logical structure. As a result, rudimentary production equipment and the low technical content of small- and medium-sized enterprises in the renewable resource recycling industry have led to serious problems, such as high carbon emissions and ecological environmental pollution. (d) There is a low legislative support effect in PR and ESD via low levels of law and judicial interpretations (JI) and administrative regulations (AR), which can be deemed as a hindrance to low-carbon economic development. For example, the legislative support effect at the JI and AR levels is only 0.23, which is too weak to support the "double carbon" transformation.

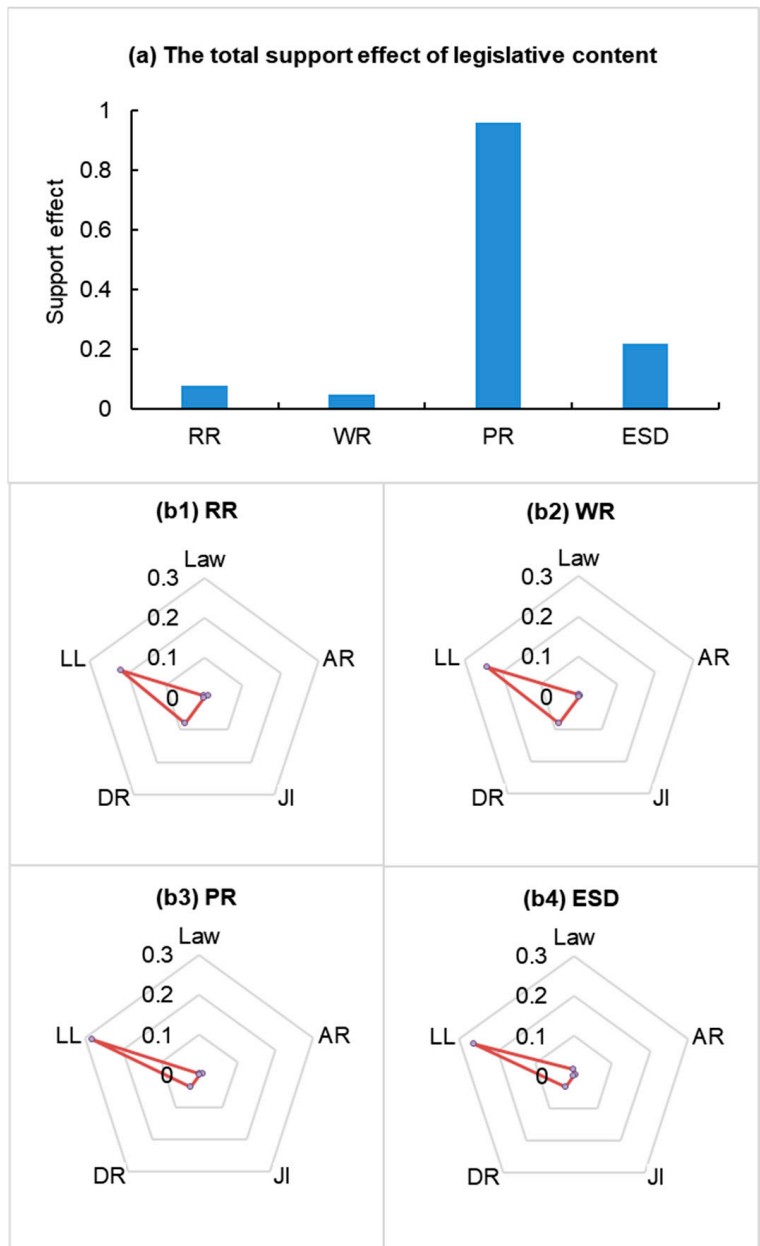

**Figure 7.** The support effect of each legislative area and corresponding legislative hierarchy (RR—resource reuse/reduction; WR—waste recycling; PR—pollution reduction; ESD—economic and social development; law—law; AR—administrative regulations; JI—judicial interpretation; DR—departmental regulations; LL—local legislation).

## *3.2. Discussion*

Based on the result analysis, various problems associated with legislative support for circular economy development under "double carbon" can be found as follows: (a) The current legislative system has paid more attention to coping with resource/environment problems after the crisis occurred, where a backward resource use pattern and emission management manner cannot match the high requirement of "double carbon". For instance, the existing laws and regulations in China pertaining to the dual-carbon circular economy predominantly focus on post-violation punitive measures, with relatively low costs of non-compliance. (b) The current legal support system cannot supply a good driving force for the transformation of production patterns and the manner of resource consumption, which can be deemed as a hindrance to circular economy development with the aim to match the

"double carbon" goal. (c) Excessive regional laws and regulations without consideration of overall/national design in the construction of the legislative system can disperse utility into various targets, which weakens the total support effect for the circular economy. For example, various aspects of circular economy development are often fragmented across different laws and regulations, lacking overall coherence and synergy. This fragmentation may result in conflicts or duplications among the relevant regulations, making it difficult to establish an organic and consistent legal framework to support the development of the circular economy. Particularly, the existing laws and regulations may ignore the requirements of resource recycling, waste management, and sustainable production and consumption patterns, which can impede the realization of the potential benefits and the transformative impact of the circular economy mode. (d) In fact, the construction of the MILC is the process of improvement from scratch. In particular, the development of a circular economy is a continuously evolving process, which can confront new changes and challenges that old laws/regulations could not cope with. Therefore, relevant new policies/laws should be enacted to adapt to these changes and fluctuations, which can improve the MILC systematically.

The corresponding suggestions to promote circular economy development can be shown as follows: (a) An advanced mindfulness of pollution control or resource planning can be fortified in the process of constructing the legislative system, which would reduce the probability of the occurrence of resource/environment problems. (b) Various legislative regulations and policies associated with technique improvement, socioeconomic development transformations, and resource use change, which would support the transformation of production patterns and the manner of resource consumption, can be intensified. For instance, by implementing effective administrative measures, such as resource allocation planning, resource pricing mechanisms, and stringent monitoring and enforcement systems, the efficient utilization of resources can be incentivized. This, in turn, contributes to the development of a circular economy by promoting resource conservation, minimizing waste generation, and facilitating the recycling and reuse of resources. (c) The government should establish an international legal support system for addressing the current low systematic degree of legalization for the circular economy to the corresponding pairs of subjects. (d) In the current "double-carbon" context, the policymakers should consider the adjustment of law and policy from different views (such as reduction, reuse, and recycling) to confront new changes and enable the coexistence of environmental and economic benefits.

In this study, the applications of the MILC combined with entropy TOPSIS method can ensure the orderly operation of the circular economy in terms of resource utilization, carbon reduction, and social development, which are worth being addressed in an international/global view, as follows: (a) The establishment of the MILC takes into account various subsystems at different levels, including resources, the environment, the economy, and society; meanwhile, it also reflects the progress of legislative efforts on the circular economy by the Chinese government from 2009 to 2022, evolving from an exploratory stage to a stage that still requires further improvement. Thus, the process of the MILC is worth promoting in countries with relatively weak rules of law to enhance the corresponding legal safeguards and ensure the sustainable development of the circular economy. (b) The "prevention or planning before incident occurrence" management manner should be fortified in the process of legislative system construction, which can reduce the loss of resource waste and the penalty of environmental pollution to a great extent, achieving a highest benefit under the "double carbon" goal. Therefore, this can be deemed as an important lesson for some developing countries that are experiencing waste/pollution and then treatment of the problem.

## 4. Conclusions

In this study, a multi-level quantitative index system of legislative support for the circular economy (MILC) through the entropy TOPSIS method was proposed for reflecting on the dynamic impact of changes in the number of rule-of-law policies at the legislative

level regarding the circular economy. It can not only express legal support for the circular economy in four areas (including resource output, waste recycling, environmental protection, and economic and social development), but also responds to the process of environmental law improvement dynamically. Compared with past studies, on the one hand, it can reflect the ecological effects and quantify the legal support for the circular economy from four aspects, which avoids the complexity of estimation processes in past studies; on the other hand, it is an effective/direct method to support the adjustment of current circular economy policies (as shown in Table 1). With the aid of the application of the MILC in China from 2009 to 2022, a number of discoveries can be discussed, as follows: (a) China's circular economy and its rule of law guarantee are still facing many challenges (e.g., the low systematic degree of legalization for industrial development, unmatched supporting legal system, and backward concepts and consciousness of circular economy legalization), which is not conducive to supporting the two-carbon target. (b) Environmental protection is more improved than other legislative areas in the four dimensions of the circular economy, which indicates that current legislative support prefers environmental governance but neglects the manner of green consumption and economic transformation in the current legislative support system construction. (c) The legal construction of the circular economy shows a trend of accelerating first and then slowing down (in 2017), which revealed diverse local policy/regulation but lacking in overall design, which reduced the efficiency of the legislative support for the circular economy in the last decade. Under these situations, various management strategies according to the identification of the importance of the legislative support system for the circular economy were addressed (strengthen the awareness of advanced pollution control or resource planning in the process of legislative system construction; strengthen laws, regulations, and policies related to technological improvement, socioeconomic development transformation, and resource utilization change; the government should establish an international legal support system; the construction of the MILC emphasizes reduction, reuse, and recycling in the circular economy, and should also incorporate elements of green and low-carbon cycles, which will not only change the pattern of resource consumption and carbon emission from the view of systematic legalization but also transfer the economic development mode to match the goal of "double carbon" through the force of legislation).

However, the legislative support of the circular economy with the goal of "double carbon" is a complicated and complex project, which is impacted by various objective and subjective factors. For instance, international cooperation and communication can improve the speed of economic transformation in an open manner, which requires a more widespread view of the legal support system to confront the challenges without limitations at the domestic level. More indicators can be incorporated into the MILC to achieve the "double carbon" goal in a global context, which may combine global climate governance, technology improvement, and environment regulation into a framework with the aim to improve its scientific level. Thus, more multivariate measures and uncertain analysis methods can be introduced to make the results more accurate. In addition, Granger causality and the entropy method were used to avoid the influence of subjective factors; however, the selection of indicators is still incomplete and the mode lacks consideration of acceptance and rejection criteria, which introduces bias to the results. These areas will be further improved upon in the subsequent studies.

**Author Contributions:** Conceptualization, R.G.; methodology, H.H.; software, X.Z. (Xinyu Zhang); investigation and data, R.G. and X.Y.; writing—original draft preparation, R.G., H.H. and X.Z. (Xinyu Zhang); revision, X.Z. (Xueting Zeng). All authors have read and agreed to the published version of the manuscript.

**Funding:** This research received no external funding.

**Institutional Review Board Statement:** Not applicable.

**Informed Consent Statement:** Not applicable.

**Data Availability Statement:** The data presented in this study are from the China National Legal and Regulatory Affairs Website (http://www.chinalaw.gov.cn, accessed on 25 May 2023), the Ministry of Ecology and Environment of the People's Republic of China (http://www.mee.gov.cn, accessed on 25 May 2023), the National Development and Reform Commission (NDRC) (http://www.ndrc.gov.cn, accessed on 25 May 2023), the Ministry of Industry and Information Technology of the People's Republic of China (http://www.miit.gov.cn, accessed on 25 May 2023), and the National Bureau of Statistics (http://www.stats.gov.cn/, accessed on 25 May 2023).

**Acknowledgments:** The authors are grateful to the editors and the anonymous reviewers for their insightful comments and suggestions.

**Conflicts of Interest:** The authors declare no conflict of interest.

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
