# Peer review of "An Analysis of Legislative Support Effect for Circular Economy Development in the Context of “Double Carbon” Goal in China"

_sustainability, doi:10.3390/su151310166_

Round 1

Reviewer 1 Report

The paper show an appropriate academic rigour. The figures are particularly informative.

The point of departure could have been expanded on. Something about the global significance of China's circular economy policy could have been added.

Systematicity could be further explained.

There is a difference between change in policies and enactment of the circular economy. If suitable policies have been put in place why should they change? This could be discussed.

The major revision is to do will spelling and word usage. The contents and research of the paper I believe reaches the required level.

Perhaps when you say 3R principle, the first time say 3R principle of waste management.

Often the word "the" is missing and would be expected. Sometimes the singular is used when the plural is needed. Sometimes the plural is used when the singular is needed (advices).

and was, not and which was

Until 2020 should be Up until 2020

ertain

0.092 in 2019 to 0.0186 in 2010 ?

Vandalism is a strange word to use.

has to pay

suppot

Reginoanl developing levle ?

An advanced mind should be An advanced mindset

Circle economy ?

Suituiations ?

Legatvie ?

messures ?

suppprot ?

currency of results - currency is a strange word to use.

Author Response

We are grateful for the reviewer’s helpful suggestions. 

Accordingly, we have responsed the comments and revised the manuscript. 

Reviewer 2 Report

Lack of Methodological Details: The provided information lacks specific details about the methodology, such as data sources, sample selection, and potential limitations. Without these details, it is challenging to assess the robustness and reliability of the findings.

Limited Generalizability: The study focuses on legislative support for the circular economy in China from 2009 to 2022. The findings may not be easily generalizable to other countries or regions with different legal frameworks and socio-economic contexts. The limited scope might restrict the applicability of the study's conclusions beyond the Chinese context.

Data Availability and Quality: The paper does not provide information about the availability and quality of the data used in the analysis. The reliability and accuracy of the findings depend on the data sources and the validity of the indicators used in constructing the multi-level quantitative index system.

Insufficient Discussion of Limitations: The weaknesses and limitations of the study are not extensively discussed in the provided sections. A thorough exploration of the study's limitations, such as potential biases, data gaps, or methodological constraints, would enhance the transparency and credibility of the research.

The content should be more succinctly described and contextualized, providing clear connections to previous research in the field.

Clearly state the research design, questions, hypotheses, and methods in a more explicit manner.

Enhance the coherence, balance, and compelling nature of the arguments and discussion of findings.

Improve the clarity of presenting empirical research results.

Ensure the article is adequately referenced and include relevant sources to support the conclusions.

The English language requires moderate editing to improve clarity and readability.

"pay more attention to cope" should be "pay more attention to coping"

"impelecaiton" should be "implementation"

"regulation impelecaiton" should be "regulatory implementation"

"concept and consciousness of circular economy legalization" should be "concepts and consciousness of circular economy legalization"

Author Response

(The authors gave the same response as above.)

Reviewer 3 Report

the suggestions are below:

1. the english wording, typo errors should be revised.

2. the authors stated that 'various results associated with total legislatvie support effect for circular economy development', the brief intro in this place about these results. then start to explain..

3. what is the accept and reject criteria for the model

4. the discussion shoul be prometed by the appropriate bibliography

5. what is the pros and cos of the applied method and the results.

6. what is the author suggestions for the obtaining, is there any policy strategy or management strategy, please explain

english, wording, typo, punctuation should be carefully checked.

Author Response

(The authors gave the same response as above.)

Reviewer 4 Report

This paper provides various insights into the impact of rule-of-law policies at the legislative level on the circular economy. However, the following improvements must be considered before publication.

1.       Numbers of references is very low and majority is very old. Kindly add more recent literature preferably from 2023 (most papers), 2022, 2021, 2020, and 2019.

2.       Novelty of the paper has not been supported by exiting literature. It is better to include a Table which would compare few preferably latest and highly relevant papers. In this Table the research need and gaps could be seen clearly.

3.       Results and discussion section could be merged. Kindly discuss each key result only but in detail.

4.       How the results are novel with respect to the existing literature. How these results can help the practitioners in a different way.

5.       A short but comprehensive flow chart is required in the Methodology section. This flow chart would explain the research steps such as data collection methods, data analysis methods, and data validation methods.

English is fine.

Author Response

(The authors gave the same response as above.)

Reviewer 5 Report

Very good article. My comments are as follows:

Introduction needs to be improved. It should include more things that highlight why the issue addressed in the article is important, you should give there the structure and purpose of the article (expand this paragraph because it is too short).

Give a review of the literature on the topic you have developed.

The methodology is not clear, please correct it.

The results are interesting, but don't link it to the Discussion section. These are two separate chapters.

Discussion should be expanded and corrected, written separately from the results.

Conclusions must be improved! They're too short. Consider whether you should move some paragraphs from the discussion to conclusions.

Present directions for further research.

Author Response

(The authors gave the same response as above.)

Round 2

Reviewer 1 Report

Some suggested English connections made below. There maybe more.

Suggested change of title "An Analysis of"

general

the estimating process

China set 33

circular economy 75

circular economy 86

circular economy 94

deemed to be holding back the circular economy development 133

were ____ further 239

law (bold ?) 241

the number 245

486 I think you mean incorporated

which ___ will 493

Author Response

RESPONSES TO REVIEWER ONE’S COMMENTS

We are grateful to Reviewer Two for his/her insightful review. The provided comments have contributed substantially to improving the paper. According to them, we have revised the manuscript, with the details explained as follows.

Point #1

COMMENT: Suggested change of title "An Analysis of"

RESPONSE: We are grateful for the reviewer’s helpful suggestions. Accordingly, we have revised the title of paper as “An Analysis of Legislative Support Effect for Circular Economy Development in the Context of “Double Carbon”goal in China”

Point #2

COMMENT: general the estimating process China set 33

circular economy 75

circular economy 86

circular economy 94

deemed to be holding back the circular economy development 133

were ____ further 239

law (bold ?) 241

the number 245

486 I think you mean incorporated

which ___ will 493

RESPONSE: We much thank for the reviewer’s useful comments. Accordingly, we have revised corresponding expressions as follows:

In 2020, China set the goals of peaking CO2 emissions by 2030 and achieving carbon neutrality by 2060 (called “double carbon” goal), which requires Chinese government to take a series of positive actions (including market access, legal improvement, industrial structure adjustment, consumption pattern optimization) to confront contradiction between human energy consumption structure and resource and environmental constraints. .(on pages 33 to 37 of the manuscript)

and

however, the conceptualized data through ecological footprint is not an effective / direct way to support adjustment of current policies of circular economy. (on pages 73 to 75 of the manuscript)

and

Various researchers have focused on qualitative analysis of legislative promulgation on circular economy, with aim to achieve the goal of "double carbon" in China [30-33]. (on pages 87 to 89 of the manuscript)

and

 It can not only express legal support for circular economy in four sights (including resource output, waste recycling, environmental protection, economic and social development), but also response the process of environmental law improvement dynamically. (on pages 96 to 99 of the manuscript)

and

The number of laws, regulations and policy documents related to circular economy development were also taken into account issued by the National Development and Reform Commission, the Ministry of Industry and Information Technology, the Ministry of Environmental Protection and other relevant departments [38-41]. (on pages 241 to 245 of the manuscript)

and

The collected policy texts, laws and regulations were cross-referenced and screened according to the conceptual scope and implementation of circular economy further. (on pages 245 to 247 of the manuscript)

and

Strengthen laws, regulations and policies related to technological improvement, socioeconomic development transformation and resource utilization change (on pages 486 to 488 of the manuscript)

and

“which will not only change the pattern of resource consumption and carbon emission from the view of systematic legalization, but also transfer economic development mode to match the goal of "double carbon" through the force of legislation."(on pages 491 to 494 of the manuscript)

Reviewer 2 Report

the abstract provides an overview of the importance of the circular economy, highlights challenges faced by China's circular economy and its legal framework, and mentions suggestions for improvement. Addressing these improvements will strengthen abstract effectiveness, attracting readers and maximizing research impact. Continuous refinement of abstract writing skills is essential for future research endeavors

Author Response

Point #1

COMMENT: the abstract provides an overview of the importance of the circular economy, highlights challenges faced by China's circular economy and its legal framework, and mentions suggestions for improvement. Addressing these improvements will strengthen abstract effectiveness, attracting readers and maximizing research impact. Continuous refinement of abstract writing skills is essential for future research endeavors.

RESPONSE: We are thankful for the reviewer’s insightful comments. According to the reviewer’s comments, we have revised abstract, where challenges faced by China's circular economy and its legal framework have been enhanced in the beginning of abstract; then suggestions and essential for future research endeavors have been added in the end of abstract. The revised sections are shown as follows:

“In the requirement of "double carbon" goal, China has confronted lack of driving force for the low-carbon transformation of socioeconomic development, which requires a comprehensive law and strategy support system for supporting circle economy. In this study, a framework associate with a multi-level quantitative index system associated with legislative support for circular economy (MILC) through entropy TOPSIS method has been developed. It can not only reflect legal support for circle economy in four-content sights based on “3R” principle, but also response the process of environmental law improvement dynamically. The legislative support effect can be applied and analyzed in China's circular economy for the period from 2009 to 2022, which can response the process of legal improvement on the environment dynamically. The obtained results show that China's circular economy and its rule of law guarantee system are still facing many challenges, such as the low systematic degree of legalization for industrial development, unmatched supporting legal system and backward concepts and consciousness of circular economy legalization. Various suggestions according to the identification of importance of legal support system for circular economy have been obtained, which can not only encourage to reduce resource consumption and carbon reduction from the view of systematic legalization, but also promote socioeconomic transformation to match the goal of "double carbon".” (On pages 10 to 27 of the revised manuscript)

Reviewer 3 Report

The authors appropriately revised the MS.

Author Response

Thank you for reviewer 3's suggestions. We will express our gratitude in the acknowledgments.